# Association Between Higher Body Mass Index and the Risk of Lumbar Spinal Stenosis in Korean Populations: A Nationwide Cohort Study

**DOI:** 10.3390/jcm13237397

**Published:** 2024-12-04

**Authors:** Ji-Hyun Ryu, Kyungdo Han, Ju-Yeong Kim

**Affiliations:** 1Department of Orthopedic Surgery, Yeouido St. Mary’s Hospital, College of Medicine, The Catholic University of Korea, Seoul 07345, Republic of Korea; ziggy777@hanmail.net; 2Department of Statistics and Actuarial Science, Soongsil University, Seoul 06978, Republic of Korea; 3Department of Orthopedic Surgery, Gyeongsang National University Changwon Hospital, Gyeongsang National University School of Medicine, Changwon 51472, Republic of Korea

**Keywords:** spinal stenosis, body mass index, cohort study, obesity, risk factors, Korean population

## Abstract

**Background/Objectives:** Despite the increasing prevalence of both spinal stenosis and obesity, their association remains controversial. This study aimed to investigate the relationship between body mass index (BMI) and the risk of lumbar spinal stenosis in the Korean population using nationwide data. **Methods:** We analyzed data from 2,161,684 adults aged ≥40 years who underwent health examinations in 2009 using the Korean National Health Insurance System database. Participants were categorized by BMI into five groups: underweight (<18.5), normal weight (18.5–22.9), overweight (23.0–24.9), obesity class I (25.0–29.9), and obesity class II and above (≥30). Cox proportional hazards models were used to evaluate the association between BMI and lumbar spinal stenosis risk, adjusting for demographic characteristics, lifestyle factors, and comorbidities. **Results:** During the 10-year follow-up period, the incidence rate of lumbar spinal stenosis increased progressively with higher BMI categories, from 32.77 per 1000 person-years in the underweight group to 51.51 in the obesity class II and above group. In the fully adjusted model, compared to the normal weight group, the hazard ratios (95% confidence intervals) were 0.801 (0.787–0.815) for underweight, 1.132 (1.126–1.139) for overweight, 1.245 (1.238–1.252) for obesity class I, and 1.348 (1.331–1.366) for obesity class II and above. The association was stronger in females and participants aged <65 years. **Conclusions**: A higher BMI was independently associated with an increased risk of lumbar spinal stenosis in the Korean population. This association remained robust after adjusting for various confounding factors, suggesting BMI as a significant risk factor for spinal stenosis.

## 1. Introduction

Spinal stenosis is a condition in which the spinal canal narrows, leading to the compression of the spinal cord and nerves. This compression results in characteristic symptoms such as pain, extremity numbness, and intermittent claudication during prolonged standing or walking [1]. The prevalence of spinal stenosis is increasing, and it adversely affects quality of life [2].

The pathogenesis of lumbar spinal stenosis involves degenerative processes of the spine, leading to the progressive compression of neural structures within the spinal canal. The causes can be congenital or acquired, with acquired causes being more common in the elderly population [3]. The onset and progression of spinal stenosis are currently known to be influenced by various factors, and associations with other diseases have been identified through numerous studies. Known factors to date include age [4], abnormal physical loading [5], genetic factors [6], smoking [7], metabolic syndrome [8], stress and psychiatric disorders [9], and regular exercise [10]. Various studies are also being conducted to investigate the relationships with other factors.

In recent years, the number of individuals suffering from obesity has increased globally due to changes in dietary habits, lack of physical activity, and rising stress levels [11]. This phenomenon has been observed not only in Western countries, but also in Asian countries [12]. Obesity is becoming increasingly common among patients with lumbar spinal stenosis [13]. Medical complications associated with obesity, including metabolic diseases and musculoskeletal issues, can be extensive and significantly limit an individual’s functional capacity and participation in daily life [14].

Recent studies report that lifestyle-related diseases, such as diabetes mellitus, peripheral artery disease, and heart disease, are frequently associated with symptomatic lumbar spinal stenosis [9,15,16]. However, despite the intuitive assumption that body mass index (BMI) and obesity may influence the development of lumbar spinal stenosis, relatively few studies have investigated the association between BMI and lumbar spinal stenosis. Some studies have suggested that a high BMI, particularly being overweight or obese, may be associated with disc degeneration [17] and spinal stenosis. However, this association remains largely speculative, with several studies presenting strong evidence against it [18,19]. These varied results make it challenging to draw a definitive conclusion.

Related studies are primarily conducted through retrospective analyses utilizing accumulated data. However, studies addressing the association between BMI and lumbar spinal stenosis through nationwide big data analysis are highly limited, with only a few including BMI among various risk factors. To our knowledge, this is the first nationwide big data analysis on the association between BMI and the risk of lumbar spinal stenosis in the Korean population. Therefore, this study aims to clarify the correlation between BMI and the risk of lumbar spinal stenosis by analyzing nationwide data from South Korea.

## 2. Methods

### 2.1. Data Source and Study Population

We used the Korean National Health Insurance System (NHIS) database. This database is part of a government-run social health insurance system, which mandates most of the population to enroll and requires individuals aged 40 and older to undergo health checkups every 1–2 years. The study population was selected, and data were collected through the health examination database and claims datasets from the NHIS.

The NHIS claims dataset is structured based on the International Classification of Diseases, Tenth Revision (ICD-10), providing examination and treatment information suitable for population-based cohort studies [20]. This study focused on individuals aged 40 and older who underwent health checkups in 2009. Among the study population, patients with spinal stenosis were identified using the NHIS claims dataset, with the diagnosis based on the ICD-10 code M48.06, which specifically refers to lumbar spinal stenosis. The diagnosis of lumbar spinal stenosis in the Korean National Health Insurance System is primarily based on patient history, neurological examination, and imaging studies such as computed tomography (CT) and magnetic resonance imaging (MRI). Key symptoms include neurogenic claudication, sensory abnormalities, and motor deficits. These clinical findings are complemented by imaging-confirmed morphological or pathophysiological changes. All diagnostic codes are assigned by board-certified physicians following thorough clinical evaluations and imaging studies.

Patients diagnosed with lumbar spinal stenosis before 2009 or those with missing data were excluded. Additionally, patients diagnosed with lumbar spinal stenosis within one year of their health examination date were excluded to avoid potential reverse causality, as a diagnosis within one year of the examination may not allow for the establishment of causality. A total of 2,161,684 individuals were finally included in this cohort study (Figure 1).

The cohort was followed from the time of lumbar spinal stenosis diagnosis in the NHIS claims dataset or until the end of the follow-up period (31 December 2020).

### 2.2. Ethics

The study protocol received review and approval from our hospital’s Institutional Review Board (IRB) and was conducted in accordance with the principles of the Declaration of Helsinki. Permission was obtained from the Korean NHIS to evaluate information from the NHIS database. Due to the anonymized nature of the data, the requirement for informed consent was waived.

### 2.3. Data Collection and Comorbidities

Health examinations consisted of self-reported questionnaire responses and direct clinical measurements. Lifestyle information, including smoking, alcohol consumption, physical activity, and income level, was collected through the questionnaire, while height, weight, and blood pressure measurements were performed by healthcare professionals. Height was measured without shoes, and weight in light clothing. Body mass index (BMI) was calculated as weight (kg) divided by height squared (m^2^). Blood samples were collected after an 8 h fast.

To account for potential effects of comorbidities, hypertension (HTN), diabetes mellitus (DM), dyslipidemia and chronic kidney disease (CKD) were defined according to specific criteria. HTN was identified by ICD-10 codes I10–13 or I15 and was confirmed if at least one antihypertensive medication was prescribed or if systolic/diastolic blood pressure was ≥140/90 mmHg. DM was defined by ICD-10 codes E11–14 and at least one antidiabetic medication prescription or a fasting blood glucose level of ≥126 mg/dL. Dyslipidemia was defined by ICD-10 code E78, along with a prescription for lipid-lowering agents or a fasting total cholesterol level of ≥240 mg/dL. CKD was defined by ICD-10 codes N18 and a glomerular filtration rate (GFR) below 60 mL/min/1.73 m^2^ for at least three months.

### 2.4. Statistical Analysis

#### 2.4.1. Basic Statistical Analysis

Baseline characteristics were summarized as means and standard deviations for continuous variables and as counts and percentages for categorical variables. Continuous variables were compared using analysis of variance (ANOVA), while categorical variables were compared using the chi-square test. The incidence rate (IR) of lumbar spinal stenosis was calculated as the number of cases per 1000 person-years over the total follow-up period, with each BMI category provided with its respective IR.

All statistical analyses were performed using SAS software version 9.4 (SAS Institute, Cary, NC, USA) with a significance level set at *p* < 0.05. Visualizations of the results were generated using Python 3.10.12 and the matplotlib library version 3.7.1.

#### 2.4.2. Analysis of BMI Categories and Cox Proportional Hazards Model

Cox proportional hazards models were employed to evaluate the association between BMI and the risk of lumbar spinal stenosis. This model was chosen because it allows for the analysis of time-dependent risks, accommodating censored data due to loss of follow-up or mortality during the observation period. By incorporating time-to-event data, Cox models ensure a robust and accurate estimation of hazard ratios across BMI categories.

Participants were categorized by BMI into five groups: underweight (<18.5), normal weight (18.5–22.9), overweight (23.0–24.9), obesity class I (25.0–29.9), and obesity class II and above (≥30). The total number of participants and event cases were presented for each BMI group. The association between BMI and the risk of lumbar spinal stenosis was analyzed using Cox proportional hazards models. Models 1 through 3 adjusted for the following variables: Model 1 provided unadjusted estimates of the association between BMI and incident cases. Model 2 adjusted for age and sex. Model 3 further adjusted for income level, smoking status, alcohol consumption, regular exercise, DM, HTN, dyslipidemia, and CKD. Hazard ratios (HRs) and 95% confidence intervals (CIs) were calculated for each BMI category, with the normal weight group (BMI 18.5–22.9) set as the reference group.

#### 2.4.3. Analyses of Detailed BMI Classification and Subgroup Effects

Additionally, for a more detailed analysis of the relationship between BMI and lumbar spinal stenosis risk, BMI was divided into 18 categories. These BMI categories were further subdivided as follows: <17, 18.0–18.9, 19.0–19.9, 20.0–20.9, 21.0–21.9, 22.0–22.9, 23.0–23.9, 24.0–24.9, 25.0–25.9, 26.0–26.9, 27.0–27.9, 28.0–28.9, 29.0–29.9, 30.0–30.9, 31.0–31.9, 32.0–32.9, 33.0–33.9, and ≥35. HRs and 95% CIs were estimated for each BMI category using Cox proportional hazards models. The same three models were applied as in the previous analysis, with BMI 21.0–21.9 as the reference category. To examine whether the association between obesity and lumbar spinal stenosis varied across different demographic characteristics, subgroup analyses were conducted. Subgroup analyses assessed the relationship between BMI and each subgroup stratified by obesity status.

Subgroups were stratified by age (<65 years, ≥65 years), sex (male, female), income level (Q1, Q2–4), abdominal obesity (yes, no), smoking status (nonsmokers/former smokers, current smokers), alcohol consumption (non-drinker/light drinker, heavy drinker), regular exercise (yes, no), DM (yes, no), HTN (yes, no), dyslipidemia (yes, no), and CKD (yes, no). Income level was categorized based on health insurance premiums, with the lowest 25% (Q1) and others (Q2–4) as comparison groups. For each subgroup, non-obese individuals served as the reference (Ref) group, and HRs with 95% CIs for obese individuals were calculated. The final model adjusted for age, sex, income, smoking, drinking, regular exercise, DM, HTN, dyslipidemia, and CKD. Interaction tests were conducted to assess differences in the impact of obesity on lumbar spinal stenosis across subgroups. An interaction *p*-value of <0.05 indicated a statistically significant difference in the effect of obesity on lumbar spinal stenosis risk within the respective subgroups.

## 3. Results

### 3.1. Baseline Characteristics

A total of 2,161,684 study participants were divided into five BMI categories. The normal weight group (BMI 18.5–22.9) comprised the largest proportion with 803,691 individuals (37.2%), followed by the obesity class I group (BMI 25.0–29.9) with 665,019 individuals (30.8%), the overweight group (BMI 23.0–24.9) with 574,882 individuals (26.6%), the obesity class II and above group (BMI ≥ 30) with 67,354 individuals (3.1%), and the underweight group (BMI < 18.5) with 50,738 individuals (2.3%).

In terms of demographic characteristics, participants aged 65 and older were most represented in the underweight group at 24.87%, compared to approximately 13–14% in other BMI categories (*p* < 0.001). Gender distribution also differed significantly across BMI categories (*p* < 0.001); the proportion of females was highest in the underweight and normal weight groups at 52.46% and 53.26%, respectively, while males were more prevalent in the overweight and obesity class I groups, accounting for 56.91% and 60.81%, respectively. The prevalence of chronic diseases showed a distinct increasing trend with higher BMI categories. The prevalence of HTN increased from 17.7% in the underweight group to 57.63% in the obesity class II and above group, DM from 6.82% to 21.6%, and dyslipidemia from 9.51% to 34.64%. Regarding lifestyle factors, the current smoking rate was highest in the underweight group at 28.47%, and heavy drinking was most prevalent in the obesity class I group at 9.67%. Regular exercise was most common in the overweight group (21.55%) and least common in the underweight group (13.79%). Anthropometric and clinical examination results showed that, as BMI increased, waist circumference, systolic and diastolic blood pressure, fasting glucose, total cholesterol, and triglycerides gradually increased, while HDL-cholesterol tended to decrease (Table 1).

### 3.2. Spinal Stenosis Risk by BMI Categories

The association between BMI and the risk of lumbar spinal stenosis was analyzed using five BMI categories. The IR of lumbar spinal stenosis showed a gradual increase with higher BMI categories, with the underweight group (<18.5) having an IR of 32.77 and the obesity class II and above group (≥30) having an IR of 51.51 (Table 2).

In Model 1, which was unadjusted, the risk of lumbar spinal stenosis showed a progressive increase with higher BMI. Compared to the normal weight group, the HR for the underweight group was 0.872 (95% CI: 0.857–0.887), indicating the lowest risk, while the obesity class II and above group had the highest HR at 1.372 (95% CI: 1.355–1.389).

The age- and sex-adjusted Model 2 showed similar trends. The HR for the underweight group decreased further to 0.799 (95% CI: 0.785–0.813), while the obesity class II and above group still showed a high HR of 1.367 (95% CI: 1.350–1.384).

In the fully adjusted Model 3, the dose–response relationship between BMI and lumbar spinal stenosis risk remained. Compared to the normal weight group, the HR for the overweight group was 1.132 (95% CI: 1.126–1.139), obesity class I group was 1.245 (95% CI: 1.238–1.252), and obesity class II and above group was 1.348 (95% CI: 1.331–1.366). In contrast, the HR for the underweight group was 0.801 (95% CI: 0.787–0.815), indicating approximately 20% lower incidence compared to the normal weight group.

Kaplan–Meier estimates demonstrated an increase in cumulative incidence probability over the 10-year follow-up period across all BMI categories, with higher BMI groups showing steeper increases in lumbar spinal stenosis incidence over time. The obesity class II and above group (BMI ≥ 30) exhibited the highest cumulative incidence, while the underweight group (BMI < 18.5) had the lowest, consistent with the results from the Cox proportional hazards model. The risk ratio increased with higher BMI across all three adjustment models. With BMI 18.5–22.9 as the reference (HR = 1), the HR was highest in the obesity class II and above group (BMI ≥ 30) across all models, particularly in Model 3 (HR = 1.348). Conversely, the underweight group (BMI < 18.5) consistently showed a lower risk than the reference group (Model 3 HR = 0.801). This linear relationship visually demonstrates the dose–response relationship between BMI and the risk of lumbar spinal stenosis (Figure 2).

To further examine the relationship between BMI and lumbar spinal stenosis risk, BMI categories were further subdivided for a more detailed analysis.

### 3.3. Detailed BMI Category Analysis

To further evaluate the relationship between BMI and the risk of lumbar spinal stenosis, BMI was divided into 19 specific subcategories as presented in Table 3. This detailed BMI subcategory analysis corroborated the progressive increase in lumbar spinal stenosis risk associated with higher BMI levels.

In the unadjusted Model 1, the HR for the BMI < 17 category was 0.863 (95% CI: 0.843–0.883), which represented the lowest lumbar spinal stenosis risk across all categories. HRs progressively increased with higher BMI, reaching 1.477 (95% CI: 1.414–1.543) in the BMI ≥ 35 category. The age- and sex-adjusted Model 2 maintained this trend. The HR for the <17 BMI category decreased to 0.766 (95% CI: 0.749–0.784), while the HR for the BMI ≥ 35 category was 1.415 (95% CI: 1.355–1.479). In the fully adjusted Model 3, the dose–response relationship between BMI and lumbar spinal stenosis risk was also observed. Compared to the reference category (BMI 21.0–21.9), the HR for BMI 25.0–25.9 was 1.190 (95% CI: 1.179–1.202), and BMI 30.0–30.9 had an HR of 1.324 (95% CI: 1.298–1.350). Notably, the HR for the BMI 32.0–32.9 category was 1.328 (95% CI: 1.288–1.369), with even higher HRs for categories above 35, peaking at an HR of 1.400 (95% CI: 1.340–1.462), indicating a significantly elevated lumbar spinal stenosis risk in cases of severe obesity. In the underweight range, a gradual decrease in lumbar spinal stenosis risk was observed with lower BMI values. The HR for BMI 18.0–18.9 was 0.828 (95% CI: 0.813–0.844), while the <17 category exhibited the lowest risk, with an HR of 0.767 (95% CI: 0.750–0.785).

To determine if the relationship between BMI and lumbar spinal stenosis risk differed by population characteristics, a subgroup analysis was conducted.

### 3.4. Subgroup Analysis

The subgroup analysis demonstrated significant effect modification by age, sex, abdominal obesity, smoking status, alcohol consumption, regular exercise, diabetes, and dyslipidemia, with interaction *p*-values < 0.01 across these factors (Table 4).

In the analysis stratified by sex, the association between obesity and lumbar spinal stenosis was found to be stronger in females (HR, 1.245; 95% CI, 1.237–1.253) compared to males (HR, 1.142; 95% CI, 1.134–1.150), with an interaction *p*-value of <0.001. Similarly, the age-stratified analysis showed that participants aged <65 years had a more pronounced association (HR, 1.202; 95% CI, 1.195–1.208) than those aged ≥65 years (HR, 1.162; 95% CI, 1.150–1.174).

For lifestyle factors, the association between obesity and lumbar spinal stenosis was more pronounced among nonsmokers/former smokers (HR, 1.214; 95% CI, 1.207–1.220) compared to current smokers (HR, 1.121; 95% CI, 1.109–1.134), with an interaction *p*-value of <0.001. Additionally, the analysis based on alcohol consumption indicated that non-drinkers and light drinkers had a stronger association (HR, 1.201; 95% CI, 1.195–1.207) compared to heavy drinkers (HR, 1.132; 95% CI, 1.112–1.151).

Among participants without abdominal obesity, the association was more substantial (HR, 1.162; 95% CI, 1.155–1.170) compared to those with abdominal obesity (HR, 1.130; 95% CI, 1.116–1.144). The regular exercise analysis also showed significant differences.

In terms of comorbidities, significant interactions were observed for DM and dyslipidemia, with a stronger association among participants without these conditions. In contrast, the association between obesity and lumbar spinal stenosis was consistent across income levels, and no significant differences were observed with respect to HTN or CKD status.

## 4. Discussion

Spinal stenosis is a degenerative disease that generally affects adults in their sixth and seventh decades of life [21]. Among these, acquired spinal stenosis is known to increase in prevalence with age. The estimated prevalence of symptomatic spinal stenosis in the general population ranges from 8.4% [22] to 9.3% [23] and is rising worldwide [24]. As many countries, including those in Asia, face an aging society, the significance of this disease has further increased. In addition, correlations between various factors and spinal stenosis have been identified. Known factors to date include age [4], abnormal physical loading [5], genetic factors [6], smoking [7], metabolic syndrome [8], stress and psychiatric disorders [9], and regular exercise [10].

The increase in overweight and obese populations, alongside aging, is a global concern, with prevalence rates continuing to rise across many demographic groups [25,26]. In cases of obesity, mortality from cardiovascular disease is higher, and the risks for diabetes and various specific cancers are significantly increased [27,28]. Obesity is recognized as a risk factor for the development of other diseases and is a metabolic disorder that should be managed to reduce additional complications rather than for its own sake. With the increasing prevalence, importance, and clinical significance of these two conditions, we became interested in further investigating them. The authors conducted this study to clarify the association between body mass index and lumbar spinal stenosis.

The use of Cox proportional hazards models strengthened the reliability of this study by accounting for time-dependent variables and censored cases, such as participants lost to follow-up or deceased participants. By leveraging this model, we were able to identify a robust dose–response relationship between BMI and the long-term risk of lumbar spinal stenosis, providing nuanced insights into the progression of this condition over a decade.

In this nationwide big data study analyzing the relationship between BMI and the risk of lumbar spinal stenosis, we observed a clear trend indicating that the risk of incidence significantly increases with a higher BMI. Notably, this relationship persisted even in Model 3, which adjusted for various confounding variables such as age, sex, lifestyle, and comorbidities, suggesting that BMI may serve as an independent risk factor for lumbar spinal stenosis. A notable point in the comparison of Models 1, 2, and 3 is that, despite the stepwise addition of adjustment variables, there was minimal change in the hazard ratio. All models consistently demonstrated a correlation between BMI and the risk of developing lumbar spinal stenosis, indicating that the association between BMI and lumbar spinal stenosis risk remains robust even when accounting for the influence of various confounding variables. Additionally, the results of the subgroup analysis suggest that the association between BMI and lumbar spinal stenosis may vary based on population characteristics. Notably, differences were observed according to factors such as gender, age, and lifestyle, indicating that the impact of obesity on spinal stenosis may differ depending on individual characteristics. Taken together, these findings suggest that BMI can be considered a significant risk factor for lumbar spinal stenosis. To our knowledge, this is the first nationwide big data study investigating the relationship between body mass index and lumbar spinal stenosis. In studying comorbidities, our research design targeted a large general population and conducted analyses accounting for various confounding variables potentially related to spinal stenosis, making our findings more reliable than those of previous studies.

Spinal stenosis can be broadly categorized into congenital and acquired types, with this study focusing primarily on acquired spinal stenosis. The increased mechanical load associated with a higher BMI contributes to accelerated degenerative changes in lumbar structures, including facet joints and intervertebral discs. Additionally, metabolic dysfunction linked to obesity may exacerbate systemic inflammation, influencing ligamentous hypertrophy and other spinal changes.

The authors interpret the mechanisms by which a high body mass index (BMI) increases the risk of lumbar spinal stenosis as follows. Primarily, the development of spinal stenosis can be attributed to mechanical loading. Increased BMI places additional load on the spine, which may exacerbate degenerative changes and the progression of arthritis [29].

Additionally, past clinical and experimental studies suggest that obesity involves not only mechanical pathways that contribute to the development of facet joint osteoarthritis, disc degeneration, and the hypertrophy of spinal ligaments, but also metabolic obesity-specific pathways [30,31]. These degenerative changes in the spine can lead to the narrowing of the spinal canal, progressing to spinal stenosis. The fact that obese individuals are at an increased risk of developing osteoarthritis in both load-bearing and non-load-bearing joints supports this mechanism [32,33]. In addition to the direct biomechanical effects on cartilage and bone, indirect effects due to changes in body mass may be mediated by mechanoreceptors, cytokines, and growth factors. These factors can alter the properties of bone matrix, ligamentum flavum, synovium, and cartilage, thereby promoting the development of osteoarthritis, ligamentum flavum hypertrophy, and disc degeneration [17,34]. Decreased muscle mass is associated with insulin resistance, which further weakens skeletal muscles and promotes systemic inflammation [35,36]. Adiponectin and leptin, hormones secreted by adipocytes, regulate low-grade inflammation caused by obesity. Increased levels of C-reactive protein, interleukins, and tumor necrosis factors are associated with the progression of spondylosis [37,38,39]. In addition, high serum concentrations of free fatty acids are known to increase systemic inflammation and the risk of developing osteoarthritis [36,40,41]. Atherosclerosis resulting from hyperlipidemia has been proposed as a cause of disc degeneration and ischemic pain [42,43]. Lastly, obesity is associated with reduced walking capacity and kinesiophobia, both of which are known to contribute to increased muscle loss and pain [44,45,46].

Our study has several limitations. First, reverse causation may be a potential issue. To minimize this, we implemented a washout period and excluded subjects diagnosed with lumbar spinal stenosis within one year of their health screening date. This one-year period was determined based on the robust data collection and tracking capabilities of the Korean National Health Insurance System (NHIS), which systematically records medical information for approximately 97% of the population. Previous studies have shown that, even with shorter washout periods, reverse causation can be effectively minimized when data collection is comprehensive and systematic [47]. While a washout period longer than two years might further reduce reverse causality, it would also decrease the sample size and compromise statistical power [48]. Given the comprehensive nature of NHIS data and the study’s objectives, the one-year period was deemed sufficient to address reverse causality while preserving analytical validity. Second, using NHIS data limited our access to radiological information. Additionally, we lacked access to detailed medical data, which meant that factors such as trauma history or specific underlying conditions could not be included in our analysis. Third, reflecting socioeconomic status posed challenges, possibly leading to the insufficient assessment of dietary and labor-related factors. Individuals from lower socioeconomic backgrounds may experience poorer nutrition, tend toward a lower BMI, and are more likely to engage in physically demanding labor, which could increase the risk of lumbar spinal stenosis. This study lacked specific data on occupational physical activity levels due to the limitations of the NHIS dataset, which does not include occupational information to protect personal privacy. Consequently, we could not directly compare physical laborers with office workers. Instead, income level and regular exercise were included as proxy variables to account for potential disparities in physical activity.

Additionally, the purpose of this study was to investigate the relationship between BMI and the occurrence of lumbar spinal stenosis. In the Korean National Health Insurance System, diagnosis codes are assigned by board-certified physicians after clinical evaluations and imaging studies, ensuring high diagnostic accuracy. However, treatment codes for diseases, including those for medications, procedures, or surgeries, are highly diverse and complex. This complexity presents significant challenges in accurately counting the number of patients undergoing procedures or surgeries, especially in large-scale big data studies. Considering these factors might enhance the value of this study. To minimize this limitation, we adjusted for income information from health examination surveys.

In conclusion, this study demonstrates that an increase in BMI is associated with a higher risk of developing lumbar spinal stenosis in the Korean population. Although there were slight variations based on gender and age, the overall trend consistently showed an elevated risk of lumbar spinal stenosis with higher BMI levels. The analysis was adjusted for age, gender, income, smoking, alcohol consumption, physical activity, hypertension, diabetes, and dyslipidemia. Given the rising prevalence of obesity and lumbar spinal stenosis, our findings highlight the importance of weight management as a preventive measure. Public health initiatives aimed at controlling BMI may reduce the incidence of lumbar spinal stenosis and its associated socioeconomic burden. These findings differ distinctly from previous studies and provide high reliability due to the large sample size. However, a more comprehensive nationwide study based on big data and involving diverse ethnic groups is needed to clarify the generalizability and potential racial differences in the association between BMI and lumbar spinal stenosis.

## Figures and Tables

**Figure 1 jcm-13-07397-f001:**
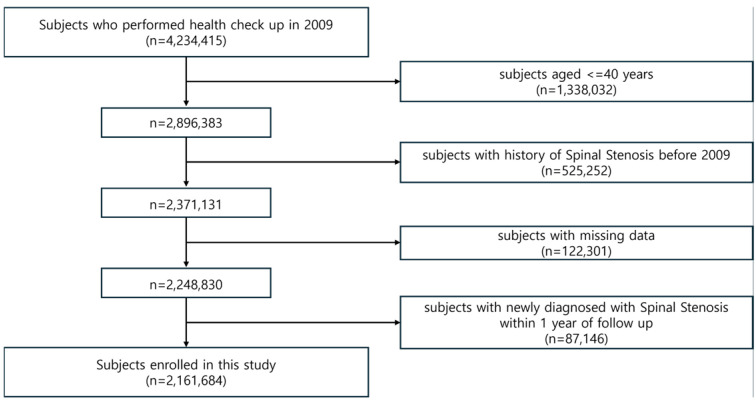
Flow chart of cohort selection.

**Figure 2 jcm-13-07397-f002:**
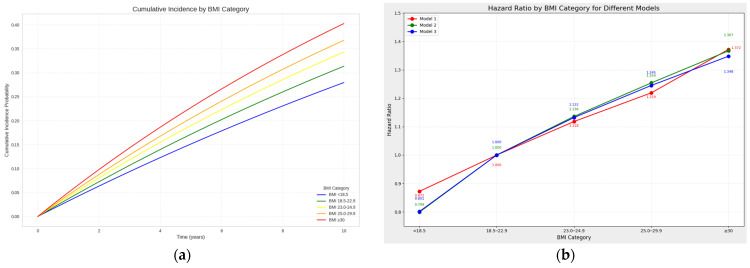
Analysis of spinal stenosis risk according to BMI categories. (**a**) Estimated cumulative incidence probabilities over a 10-year follow-up period, with each line representing a different BMI category. (**b**) Hazard ratios for spinal stenosis across BMI categories, with BMI 18.5–22.9 as reference (HR = 1.0) and error bars indicating 95% confidence intervals.

**Table 1 jcm-13-07397-t001:** Baseline characteristics of study participants according to BMI categories.

	BMI Level					
	<18.5	18.5–22.9	23.0–24.9	25.0–29.9	≥30	*p*-Value
	50,738	803,691	574,882	665,019	67,354	
Age, ≥65 years	12,617 (24.87)	112,562 (14.01)	80,480 (14)	93,954 (14.13)	9018 (13.39)	<0.0001
Sex						
Male	24,123 (47.54)	375,607 (46.74)	327,152 (56.91)	404,400 (60.81)	32,652 (48.48)	<0.0001
Female	26,615 (52.46)	428,084 (53.26)	247,730 (43.09)	260,619 (39.19)	34,702 (51.52)	<0.0001
Income, Lowest Q1	11,395 (22.46)	171,209 (21.3)	112,350 (19.54)	128,266 (19.29)	14,417 (21.4)	<0.0001
Smoking						
Non	31,425 (61.94)	520,072 (64.71)	341,419 (59.39)	377,790 (56.81)	43,155 (64.07)	<0.0001
Ex	4867 (9.59)	102,052 (12.7)	103,536 (18.01)	134,275 (20.19)	10,547 (15.66)	<0.0001
Current	14,446 (28.47)	181,567 (22.59)	129,927 (22.6)	15,2954 (23)	13,652 (20.27)	<0.0001
Drinking						
Non	32,493 (64.04)	471,583 (58.68)	309,541 (53.84)	346,741 (52.14)	39,579 (58.76)	<0.0001
Mild	15,140 (29.84)	281,202 (34.99)	219,693 (38.22)	253,938 (38.19)	21,345 (31.69)	<0.0001
Heavy	3105 (6.12)	50,906 (6.33)	45,648 (7.94)	64,340 (9.67)	6430 (9.55)	<0.0001
Regular exercise	6999 (13.79)	152,185 (18.94)	123,911 (21.55)	141,971 (21.35)	12,315 (18.28)	<0.0001
DM	3458 (6.82)	60,460 (7.52)	61,833 (10.76)	97,161 (14.61)	14,550 (21.6)	<0.0001
HTN	8982 (17.7)	170,188 (21.18)	176,144 (30.64)	274,993 (41.35)	38,818 (57.63)	<0.0001
Dyslipidemia	4824 (9.51)	121,287 (15.09)	124,586 (21.67)	181,243 (27.25)	23,333 (34.64)	<0.0001
CKD	3585 (7.07)	53,428 (6.65)	43,673 (7.6)	55,183 (8.3)	6257 (9.29)	<0.0001
Age, years	54.93 ± 12.64	52.31 ± 10.11	53.06 ± 9.63	53.18 ± 9.57	52.48 ± 9.72	<0.0001
Height, cm	161.47 ± 8.47	161.82 ± 8.25	162.8 ± 8.63	163.16 ± 8.92	161.45 ± 9.6	<0.0001
Weight, kg	45.91 ± 5.32	55.84 ± 6.55	63.62 ± 6.93	71.26 ± 8.41	83.15 ± 10.53	<0.0001
BMI, kg/m^2^	17.56 ± 0.83	21.27 ± 1.16	23.93 ± 0.57	26.69 ± 1.28	31.82 ± 2.11	<0.0001
Waist circumference, cm	67.8 ± 5.93	75.06 ± 6.18	81.4 ± 5.71	87.25 ± 6.21	96.44 ± 7.61	<0.0001
Systolic BP, mmHg	117.6 ± 15.92	120.33 ± 15.15	124.16 ± 14.91	127.47 ± 14.98	131.95 ± 15.79	<0.0001
Diastolic BP, mmHg	73.4 ± 10.24	74.94 ± 10.02	77.3 ± 9.98	79.47 ± 10.09	82.28 ± 10.7	<0.0001
Fasting glucose, mg/dL	95.92 ± 28.35	96.72 ± 24.07	100.05 ± 25.65	103.22 ± 27.27	108.24 ± 31.54	<0.0001
Total cholesterol, mg/dL	185.6 ± 34.9	193.9 ± 35.64	200.04 ± 36.82	203.32 ± 37.76	206.27 ± 39.37	<0.0001
HDL-C, mg/dL	62.83 ± 33.56	58.74 ± 30.26	54.73 ± 27.56	52.58 ± 27.36	51.93 ± 27.91	<0.0001
LDL-C, mg/dL	105.48 ± 39.1	113.7 ± 37.63	118.27 ± 38.59	119.1 ± 39.16	119.99 ± 40.37	<0.0001
Triglyceride, mg/dL	85.15 (84.79–85.51)	98.81 (98.69–98.92)	121.09 (120.92–121.27)	140.76 (140.57–140.94)	153.61 (152.98–154.24)	<0.0001

Data are presented as mean ± standard deviation or proportion (%); BMI: body mass index; CKD: chronic kidney disease. DM: Diabetes Mellitus; HTN: Hypertension; BP: Blood pressure; HDL: High-Density Lipoprotein; LDL: Low-Density Lipoprotein.

**Table 2 jcm-13-07397-t002:** Risk of lumbar spinal stenosis according to BMI categories.

BMI	N	Event	Duration, PY	IR, per 1000 PY	HR (95% C.I)		
					Model 1 ^1^	Model 2 ^2^	Model 3 ^3^
<18.5	50,738	13,507	412,190.92	32.7688	0.872 (0.857, 0.887)	0.799 (0.785, 0.813)	0.801 (0.787, 0.815)
18.5–22.9	803,691	253,179	6,731,895.55	37.6089	1 (Ref.)	1 (Ref.)	1 (Ref.)
23.0–24.9	574,882	199,344	4,744,056.27	42.0197	1.118 (1.111, 1.124)	1.136 (1.129, 1.142)	1.132 (1.126, 1.139)
25.0–29.9	665,019	247,075	5,392,078.62	45.8218	1.219 (1.213, 1.226)	1.254 (1.247, 1.261)	1.245 (1.238, 1.252)
≥30	67,354	27,268	529,408.48	51.5065	1.372 (1.355, 1.389)	1.367 (1.350, 1.384)	1.348 (1.331, 1.366)

BMI: body mass index; IR: incidence rate (per 1000 person-years); HR: hazard ratio; CI: confidence interval; ^1^ Model 1: unadjusted, ^2^ Model 2: adjusted for age and sex, ^3^ Model 3: adjusted for age, sex, income, smoking, drinking, regular exercise, dyslipidemia, and chronic kidney disease.

**Table 3 jcm-13-07397-t003:** Risk of lumbar spinal stenosis according to detailed BMI categories.

BMI	N	Event	Duration, PY	IR, per 1000 PY	HR (95% C.I)		
					Model 1	Model 2	Model 3
<17	30,111	7920	239,932.19	33.0093	0.863 (0.843, 0.883)	0.766 (0.749, 0.784)	0.767 (0.750, 0.785)
18.0–18.9	50,274	13,674	421,877.28	32.4123	0.847 (0.831, 0.862)	0.827 (0.812, 0.843)	0.828 (0.813, 0.844)
19.0–19.9	101,371	29,457	853,988.08	34.4935	0.901 (0.889, 0.913)	0.900 (0.888, 0.912)	0.900 (0.888, 0.913)
20.0–20.9	161,958	49,136	1,364,180.79	36.0187	0.940 (0.930, 0.951)	0.942 (0.932, 0.953)	0.943 (0.932, 0.954)
21.0–21.9	232,585	74,551	1,946,900.94	38.2921	1 (Ref.)	1 (Ref.)	1 (Ref.)
22.0–22.9	278,130	91,948	2,317,207.19	39.6805	1.036 (1.026, 1.046)	1.052 (1.041, 1.062)	1.051 (1.041, 1.061)
23.0–23.9	296,443	101,992	2,449,937.68	41.6304	1.087 (1.077, 1.098)	1.103 (1.092, 1.113)	1.101 (1.091, 1.112)
24.0–24.9	278,439	97,352	2,294,118.59	42.4355	1.109 (1.098, 1.119)	1.140 (1.129, 1.151)	1.137 (1.126, 1.148)
25.0–25.9	235,430	85,167	1,924,206.95	44.2608	1.157 (1.145, 1.168)	1.195 (1.183, 1.207)	1.190 (1.179, 1.202)
26.0–26.9	180,199	66,515	1,463,594.33	45.4463	1.188 (1.175, 1.200)	1.227 (1.214, 1.240)	1.221 (1.208, 1.234)
27.0–27.9	123,974	46,825	1,000,338	46.8092	1.223 (1.209, 1.238)	1.267 (1.252, 1.282)	1.259 (1.244, 1.274)
28.0–28.9	77,077	29,847	617,107.43	48.366	1.264 (1.248, 1.281)	1.302 (1.284, 1.319)	1.292 (1.275, 1.310)
29.0–29.9	48,339	18,721	386,831.92	48.3957	1.265 (1.245, 1.286)	1.316 (1.295, 1.337)	1.305 (1.284, 1.326)
30.0–30.9	29,393	11,628	232,806.21	49.9471	1.306 (1.281, 1.332)	1.336 (1.310, 1.362)	1.324 (1.298, 1.350)
31.0–31.9	14,824	6019	116,838.24	51.5157	1.347 (1.312, 1.383)	1.354 (1.319, 1.390)	1.340 (1.305, 1.376)
32.0–32.9	10,326	4328	80,420.21	53.8173	1.408 (1.366, 1.452)	1.342 (1.301, 1.383)	1.328 (1.288, 1.369)
33.0–33.9	4895	1940	38,527.66	50.3534	1.317 (1.259, 1.378)	1.341 (1.282, 1.403)	1.326 (1.267, 1.387)
34.0–34.9	3124	1294	24,300.07	53.2509	1.393 (1.319, 1.472)	1.371 (1.297, 1.448)	1.353 (1.280, 1.429)
≥35	4792	2059	36,516.08	56.3861	1.477 (1.414, 1.543)	1.415 (1.355, 1.479)	1.400 (1.340, 1.462)

BMI: body mass index; IR: incidence rate (per 1000 person-years); HR: hazard ratio; CI: confidence interval. Models were adjusted as in Table 2.

**Table 4 jcm-13-07397-t004:** Subgroup analyses of the association between obesity and risk of lumbar spinal stenosis.

		Obesity	N	Event	Duration	IR	HR (95% C.I)	*p* for Interaction
Age groups	<65	No	1,223,652	372,471	10,544,323.31	35.3243	1 (Ref.)	<0.0001
		Yes	629,401	219,871	5,256,853.22	41.8256	1.202 (1.195, 1.208)	
	≥65	No	205,659	93,559	1,343,819.43	69.6217	1 (Ref.)	
		Yes	102,972	54,472	664,633.88	81.9579	1.162 (1.150, 1.174)	
Sex	Male	No	726,882	200,044	6,187,922.46	32.3281	1 (Ref.)	<0.0001
		Yes	437,052	131,759	3,724,987.25	35.3717	1.142 (1.134, 1.150)	
	Female	No	702,429	265,986	5,700,220.28	46.6624	1 (Ref.)	
		Yes	295,321	142,584	2,196,499.85	64.9142	1.245 (1.237, 1.253)	
Income	Q2–4	No	1,134,357	361,613	9,489,012.67	38.1086	1 (Ref.)	0.6307
		Yes	589,690	215,761	4,801,151.62	44.9394	1.197 (1.190, 1.203)	
	Q1	No	294,954	104,417	2,399,130.07	43.5229	1 (Ref.)	
		Yes	142,683	58,582	1,120,335.48	52.2897	1.194 (1.181, 1.206)	
Abdominal obesity	No	No	1,352,033	434,929	11,302,239.78	38.4817	1 (Ref.)	<0.0001
		Yes	357,556	125,315	2,966,052.33	42.2498	1.162 (1.155, 1.170)	
	Yes	No	77,278	31,101	585,902.95	53.0822	1 (Ref.)	
		Yes	374,817	149,028	2,955,434.78	50.4251	1.130 (1.116, 1.144)	
Smoking	Non, Ex	No	1,103,371	377,413	9,107,575.92	41.4395	1 (Ref.)	<0.0001
		Yes	565,767	226,362	4,487,842.7	50.4389	1.214 (1.207, 1.220)	
	Current	No	325,940	88,617	2,780,566.81	31.8701	1 (Ref.)	
		Yes	166,606	47,981	1,433,644.4	33.4679	1.121 (1.109, 1.134)	
Drinking	Non, Mild	No	1,329,652	436,664	11,052,178.53	39.5093	1 (Ref.)	<0.0001
		Yes	661,603	252,019	5,320,662.55	47.3661	1.201 (1.195, 1.207)	
	Heavy	No	99,659	29,366	835,964.21	35.1283	1 (Ref.)	
		Yes	70,770	22,324	600,824.56	37.1556	1.132 (1.112, 1.151)	
Regular exercise	No	No	1,146,216	372,009	9,531,534.1	39.0293	1 (Ref.)	0.0061
		Yes	578,087	216,835	4,669,042.71	46.441	1.200 (1.194, 1.207)	
	Yes	No	283,095	94,021	2,356,608.63	39.8967	1 (Ref.)	
		Yes	154,286	57,508	1,252,444.39	45.9166	1.181 (1.169, 1.193)	
DM	No	No	1,303,560	421,946	10,922,212.62	38.6319	1 (Ref.)	0.0011
		Yes	620,662	230,084	5,058,632.16	45.4834	1.200 (1.194, 1.206)	
	Yes	No	125,751	44,084	965,930.12	45.6389	1 (Ref.)	
		Yes	111,711	44,259	862,854.94	51.2937	1.172 (1.156, 1.187)	
HTN	No	No	1,073,997	334,071	9,134,951.36	36.5706	1 (Ref.)	0.0546
		Yes	418,562	146,017	3,488,030.06	41.8623	1.201 (1.193, 1.208)	
	Yes	No	355,314	131,959	2,753,191.38	47.9295	1 (Ref.)	
		Yes	313,811	128,326	2,433,457.04	52.734	1.189 (1.180, 1.198)	
Dyslipidemia	No	No	1,178,614	371,943	9,893,708.69	37.5939	1 (Ref.)	0.0046
		Yes	527,797	190,414	4,322,291.81	44.0539	1.201 (1.194, 1.208)	
	Yes	No	250,697	94,087	1,994,434.05	47.1748	1 (Ref.)	
		Yes	204,576	83,929	1,599,195.29	52.482	1.182 (1.171, 1.193)	
CKD	No	No	1,328,625	431,119	11,101,547.59	38.8341	1 (Ref.)	0.2218
		Yes	670,933	249,009	5,457,433.55	45.6275	1.195 (1.189, 1.201)	
	Yes	No	100,686	34,911	786,595.15	44.3824	1 (Ref.)	
		Yes	61,440	25,334	464,053.55	54.5928	1.208 (1.188, 1.227)	

HR: hazard ratio; CI: confidence interval. All analyses were adjusted for age, sex, income, smoking, drinking, regular exercise, DM, HTN, dyslipidemia, and chronic kidney disease, except for the stratification variable itself, *p*, for interaction was calculated using likelihood ratio tests.

## Data Availability

The datasets used in this study are contained within the main article.

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
