# Peer review of "Association Between Higher Body Mass Index and the Risk of Lumbar Spinal Stenosis in Korean Populations: A Nationwide Cohort Study"

_jcm, 2024, doi:10.3390/jcm13237397_

Round 1
Reviewer 1 Report
Comments and Suggestions for Authors
The manuscript is clear and is a well organised statistical article. In short, this study shows that increased body mass index is associated with a higher risk of spinal stenosis.
However, there are a few issues that require revision or additional clarification.
1. This study included only the Korean population. Therefore, it would be more rigorous and appropriate to use the term ‘Korea’ rather than the East Asian. There are so many East Asian countries that were not included in this study.
2. The logical and causal relationship between BMI and spinal stenosis is not well written or detailed in the discussion section. In fact, spinal stenosis is divided into different types. Some of them cannot be explained by the conclusions of this paper.
3.As stated in the limitations section, in order to minimise reverse causality, subjects diagnosed with spinal stenosis within one year of the health screening date were excluded from this study. In practice, two years was more acceptable.
4.I doubt that this article is clinically relevant to the reader .
Author Response
Thank you for providing such a valuable comment. Following your suggestions, I have reviewed and made the following modifications to the manuscript.
The manuscript is clear and is a well organised statistical article. In short, this study shows that increased body mass index is associated with a higher risk of spinal stenosis.
However, there are a few issues that require revision or additional clarification.
- This study included only the Korean population. Therefore, it would be more rigorous and appropriate to use the term ‘Korea’ rather than the East Asian. There are so many East Asian countries that were not included in this study.
: We greatly appreciate the reviewer's insightful comment regarding the specificity of the study population. We fully agree that the term "Korean" more accurately represents the population analyzed in this study. Consequently, we have replaced all instances of "East Asian" with "Korean" throughout the manuscript to ensure clarity and rigor in describing the study population. This change has been implemented consistently in all relevant sections of the text.
- The logical and causal relationship between BMI and spinal stenosis is not well written or detailed in the discussion section. In fact, spinal stenosis is divided into different types. Some of them cannot be explained by the conclusions of this paper.
: Thank you for highlighting the need for a more detailed explanation of the mechanisms linking BMI to spinal stenosis. In response, we have expanded the discussion to describe both the biomechanical and metabolic pathways through which obesity can contribute to spinal stenosis. Specifically, we clarified that increased mechanical load and systemic inflammation associated with obesity may exacerbate degenerative changes in the spine.
Additionally, we explicitly acknowledged that spinal stenosis includes different types, but this study focuses exclusively on clinically diagnosed lumbar spinal stenosis cases identified using the NHIS criteria. To enhance transparency, we added the following explanation to the Methods and Discussion sections:
"The diagnosis of spinal stenosis in the Korean National Health Insurance System is primarily based on clinical evaluations, imaging studies, and diagnostic codes (M48.06). The NHIS data allowed us to reliably track patients with clinically significant spinal stenosis over time, ensuring high diagnostic accuracy."
- As stated in the limitations section, in order to minimise reverse causality, subjects diagnosed with spinal stenosis within one year of the health screening date were excluded from this study. In practice, two years was more acceptable.
: We appreciate the reviewer’s suggestion to consider a longer washout period to further minimize reverse causality. While we agree that a two-year period could offer additional rigor, we opted for a one-year washout to balance the need for temporal clarity with maintaining an adequate sample size. To address this point, we expanded the limitations section to explain the rationale for our decision. Specifically, we highlighted the structured and systematic data collection process in the NHIS, which ensures high traceability and reliability of medical records. We also cited studies indicating that shorter washout periods can effectively mitigate reverse causality in settings with comprehensive data systems.
- I doubt that this article is clinically relevant to the reader .
: We understand the reviewer’s concern regarding the clinical relevance of this study. To address this, we emphasized in the conclusion that managing BMI through weight control can serve as an actionable strategy to reduce the risk of spinal stenosis. By providing evidence-based insights into the association between BMI and spinal stenosis, this study aims to inform both clinicians and public health policymakers about preventive strategies. The revised conclusion highlights the significance of this study for clinical practice and public health initiatives.
Again, I appreciate your insightful feedback, and I have highlighted the changes made in Blue text in the manuscript.
Reviewer 2 Report
Comments and Suggestions for Authors
Authors present a korean national study on more than 2 million adults who underwent health care examination in 2009 using national health insurance system to investigate 10 years spinal stenosis risk in correlation to BMI of these patients; he incidence rate of spinal stenosis increased progressively with higher BMI categories, also in fully adjusted model, compared to normal weight population. Introduction provides sufficient data; Materials and Methods are clearly written. Model adjustment for other modifiers was also included which reduces the risk of bias. Potential issue is that physical activity in terms of job of the patients is not taken into consideration , so I suggest to include this into the model - it is different if someone is a physical worker and if other person is an informatician. Also, term "spinal canal stenosis" needs to be clearly defined - what was the definition, i.e. how were these data extracted at the end of the follow up in 2019? Did all 2 million patients got an examination in 2019? How was the correlation performed? Was spinal canal stenosis a radiological or clinical diagnosis, were these patients operated? Is this a lumbar or cervical or lumbar and cervical spinal canal stenosis - these are two different entities, and I suggest to divide both into two groups. How many patients did underwent surgery?
Author Response
Thank you for providing such a valuable comment. Following your suggestions, I have reviewed and made the following modifications to the manuscript:
Authors present a korean national study on more than 2 million adults who underwent health care examination in 2009 using national health insurance system to investigate 10 years spinal stenosis risk in correlation to BMI of these patients; he incidence rate of spinal stenosis increased progressively with higher BMI categories, also in fully adjusted model, compared to normal weight population. Introduction provides sufficient data; Materials and Methods are clearly written. Model adjustment for other modifiers was also included which reduces the risk of bias.
1. Potential issue is that physical activity in terms of job of the patients is not taken into consideration , so I suggest to include this into the model - it is different if someone is a physical worker and if other person is an informatician.
: We sincerely appreciate the reviewer's valuable suggestion regarding the inclusion of occupational physical activity. We completely agree that occupational activity plays a significant role in understanding physical exertion levels. However, due to the limitations of the NHIS dataset, occupational information is not collected to protect personal privacy. As a result, we could not directly compare physical laborers with office workers in this study.
To address this limitation, we incorporated proxy variables such as income level and regular exercise to partially account for variations in physical activity. This adjustment has been explicitly acknowledged in the revised manuscript as follows:
"This study lacked specific data on occupational physical activity levels due to the limitations of the NHIS dataset, which does not include occupational information to protect personal privacy. Consequently, we could not directly compare physical laborers with office workers. Instead, income level and regular exercise were included as proxy variables to account for potential disparities in physical activity."
We hope this explanation clarifies our approach and highlights our effort to address this limitation within the scope of the available data.
2. Also, term "spinal canal stenosis" needs to be clearly defined - what was the definition, i.e. how were these data extracted at the end of the follow up in 2019? Did all 2 million patients got an examination in 2019? How was the correlation performed? Was spinal canal stenosis a radiological or clinical diagnosis, were these patients operated?
: We deeply appreciate the reviewer's insightful questions and agree that providing additional clarity regarding the definition and data collection process for spinal canal stenosis is essential.
The diagnosis of spinal stenosis in this study was based on a combination of clinical evaluations and imaging studies, such as CT and MRI, to confirm morphological and pathological changes in the lumbar spine. Clinically, neurological examinations and patient history, including symptoms such as neurogenic claudication, were integral to the diagnostic process. All diagnosis codes (M48.06) in the NHIS system are assigned by board-certified physicians after thorough clinical and imaging evaluations, ensuring high diagnostic accuracy.
Regarding the data collection process, we would like to clarify that this study did not involve 2 million patients undergoing examinations in 2019. Instead, our study population was based on approximately 4 million individuals who underwent national health checkups in 2009. From this cohort, we excluded patients with pre-existing spinal stenosis or missing data, resulting in a final study population of approximately 2.1 million individuals. These individuals were then followed up until 2020 to identify cases of lumbar spinal stenosis using NHIS claims data. The NHIS system, which covers 97% of the Korean population, assigns unique patient codes that link health checkups with subsequent hospital visits and diagnostic codes, enabling seamless and reliable long-term follow-up.
Additionally, we clarified in the manuscript that we employed Cox proportional hazards models to account for time-dependent risks and censoring during the follow-up period. This modeling approach ensured robust estimation of the relationship between BMI and the risk of lumbar spinal stenosis.
These points have been incorporated into the revised manuscript to address your concerns, and we hope this explanation provides the necessary clarification.
3. Is this a lumbar or cervical or lumbar and cervical spinal canal stenosis - these are two different entities, and I suggest to divide both into two groups. How many patients did underwent surgery?
: We sincerely thank the reviewer for highlighting this important point. We acknowledge that the initial manuscript may have caused confusion by using broader terminology for spinal stenosis. However, we confirm that this study exclusively focused on lumbar spinal stenosis. To clarify this, we have revised the manuscript to consistently refer to lumbar spinal stenosis and specified that our analysis was based on the diagnostic code M48.06.
Regarding the question about surgery, we agree that treatment codes, including those for medications, procedures, and surgeries, are highly diverse and complex within the NHIS system. While this complexity limits our ability to accurately count the number of patients undergoing surgery, our primary focus was on investigating the relationship between BMI and the occurrence of lumbar spinal stenosis. This study design, coupled with the high diagnostic accuracy ensured by NHIS data, allows us to provide reliable findings despite these limitations.
Again, I appreciate your insightful feedback, and I have highlighted the changes made in Blue text in the manuscript.
Round 2
Reviewer 1 Report
Comments and Suggestions for Authors
Nothing.
Reviewer 2 Report
Comments and Suggestions for Authors
Sufficient response to remarks.